# Private-Public Opinion Discrepancy

**Ren Manfredi[1]☯, Andrea Guazzini [1,2]☯\*, Carla Anne Roos [3]☯, Tom Postmes[3]☯, Namkje Koudenburg[3]☯**

1 Department of Education, Languages, Intercultural Studies, Literature and Psychology, University of Florence, Florence, Italy, 2 Center for Study of Complex Dynamics (CSDC), University of Florence, Florence, Italy, 3 Department of Psychology, University of Groningen, Groningen, Netherlands

☯ These authors contributed equally to this work.
\* andrea.guazzini@unifi.it

**Data Availability Statement:** All relevant data are within the manuscript and its Supporting information files.

**Funding:** The author(s) received no specific funding for this work.

## Abstract

In many Western societies there are rising concerns about increasing polarization in public debate. However, statistics on private attitudes paint a different picture: the average attitudes in societies are more moderate and remain rather stable over time. The present paper presents an agent-based model of how such discrepancies between public opinion and private attitudes develop at the scale of micro-societies. Based on social psychological theorizing, the model distinguishes between two types of agents: a) those seeking to gain or maintain a good reputation and status, and b) those seeking to promote group harmony by reaching consensus. We characterized these different types of agents by different decision rules for either voicing their opinion or remaining silent, based on the behavior of their proximal network. Results of the model simulations show that even when the private attitudes of the agents are held constant, publicly expressed opinions can oscillate and (depending on the reputational concerns of individual actors) situations can occur in which minorities as well as majorities are silenced. We conclude that the macro-level consequences of micro-level decisions to either voice an opinion or remain silent provide a foundation for better understanding how public opinions are shaped. Moreover, we discuss the conditions under which public opinion could be considered a good representation of private attitudes in a society.

## 1 Introduction

The current political climate of many Western societies is characterized by high concerns about polarization. But although extreme political viewpoints are quite visible and audible in public debate and on social media, according to most statistics there is also a very large majority of moderates who are less audible: a silent majority. Thus, public opinion as inferred from what is most commonly expressed can be a poor gauge of attitudes that are privately held: in public debates certain subgroups are very vocal whilst others choose to remain silent.

The present paper presents an agent-based model of how such discrepancies between public opinion and private attitudes develop at the scale of micro-societies, based on individual agent decisions. Prior theorising, research and agent-based models have shown that fears of speaking

**Competing interests:** The authors have declared that no competing interests exist.

out may lead to silent majority effects [1]. The aim of the current paper is to shed new light on the curious phenomenon that a society can veer between public opinions (pro and contra an issue) even though the average of the privately held attitudes changes little over time. This oscillation of discrepancies between public opinion and aggregate private attitudes is not just of current political interest: we believe it is a key societal phenomenon we need to develop a better understanding of.

## 1.1 Oscillations of publicly expressed opinions

The phenomenon that discrepancies can occur between opinions that are publicly expressed and aggregated attitudes is well known. The pluralistic ignorance phenomenon shows that people's perceptions about the dominant views in their group can sometimes differ markedly from the aggregated attitudes of group members [2]. This may lead to a shift in public opinion when those who are in the majority stop expressing their views because they believe themselves to be in the minority–a phenomenon referred to as a Spiral of Silence [3]. As a result of this, some viewpoints can dominate public discourse, even though objectively few support those ideas.

One recent example of such oscillations in public opinion can be found in Dutch society. With respect to immigration, the country was known as one of the more tolerant in Europe [4]. But after 2000, public support for multiculturalism waned and support for restrictive policies of assimilation appeared to grow: the nation became one of Europe's more intolerant in its policies and public discourse surrounding immigration [4]. However, during this time the attitudes of Dutch people towards immigrants changed very little (and, if anything, became slightly more positive, [5]).

A similar phenomenon has occurred in the USA. People are less reluctant to openly express their support for restrictive immigration policies in recent years, culminating in the election of Donald Trump to president. But at the same time, attitudes towards immigration have changed little and have, in some respects at least, become more favorable towards immigration [6, 7]. In such societal phenomena we can see that public opinion and aggregated private attitudes can be at odds.

The discrepancies between aggregate attitudes and perceived public opinion can act as self-fulfilling prophecies, when they influence how people behave, vote and express themselves in adjacent domains [8, 9]. Moreover, public opinion is a major factor in politics when politicians play to the perceived will of the people to seek electoral gain. As a result, implicit norms that govern public behavior can be markedly at odds with aggregated private opinions.

In this paper, we will present and test a new agent-based model (ABM) to explain these oscillations which may, at times, lead to seemingly contradictory attitudes being expressed in public vs. held in private. This ABM is unique because it uses insights from small group research in order to model societal level public opinion dynamics. It is designed to show how macro-level societal phenomena can be explained on the basis of cognitive principles in conjunction with micro-level dynamics in small groups. The paper will start with a short overview of the existing literature on public opinion and private attitudes. This will be followed by a description of our model. Subsequently, we will present the results of the model simulations. We will end with a discussion of the theoretical implications of our findings.

## 1.2 Public opinion and personal networks

One common way of thinking about "public opinion" is that it is the aggregate of individual attitudes within a certain group or society [10–12]. However, people tend to not know what that aggregate is. Indeed, research on the pluralistic ignorance phenomenon has shown that

people's behavior can be at odds with their private attitudes [9, 13]. The consequence of this is that people continually have to guess what the aggregate attitude might be. And they do not do this in isolation: the question of "what we think" and "where we stand" is continually being discussed and assessed, often within small circles among intimates and within personal networks [14–16].

"Public opinion" is therefore based on interpretations of actions that are publicly displayed [17]. Accordingly, people's impression of the current public opinion can only be based on that which is heard and seen at a particular point in time. Personal networks and small groups play an important role here. Research shows that news stories are impactful especially when they are discussed and elaborated upon in conversations with family, friends or colleagues–a process that also occurs on social media [18, 19]. In their assessment of public opinion, people devote a lot of importance to the publicly stated views of close and intimate others, for example in reaction to current affairs and key events. So, in order to understand how public opinion is transformed, one should ask what it takes for an individual to speak out (or remain silent) about their personal views when in conversation with those that are close to them. The key question therefore has *two* sides: when and why do people choose to remain silent? And conversely: when and why do people choose to speak out?

### 1.3 Motives for speaking out: Reputation and consensus

What factors may influence the attitudes and behaviors that are publicly displayed? Of course a wide variety of factors play a role here, but in this paper we choose to focus on two classic motives that play a key role in small groups: individual and collective concerns.

With respect to individual-centric motives in small groups, it is well-known that in many group settings individuals operate strategically in relation to each other, especially when their actions are public and identifiable [20].

Spiral of Silence theory underlines self-protective within-group motives as playing a key role in public opinion dynamics [21, 22]. According to this theory, voicing one's opinion carries social risks such as rejection and isolation. Accordingly, people prefer to voice attitudes that are likely to be endorsed by the majority. The result is that people prefer to echo opinions they often hear.

But within groups, there are also other group dynamical factors that play a role. Theories such as Spiral of Silence implicitly assume that it is normal for people to voice their attitudes. But in most human societies, having an opinion is a privilege only permitted to those who have acquired a certain social status. In respect to the issue of silence or voice, we therefore contend that one key concern for individuals is to preserve their reputation and position within the group [23–27]. The implication is that individuals exercise their voice in order to acquire the group's respect and to be recognized as a valuable group member [28, 29]).

In this paper, we propose that within-group processes of reputation are a key factor in voicing one's attitude. We propose that a person can achieve a good reputation and a certain status within their group by expressing an opinion that is "up and coming"; by innovating in other words. Especially "early adopters" of a novel perspective may do so because they believe that they can reap social benefits with this: there is an opportunity to gain status and possibly power [30–32]. This prediction is different from Spiral of Silence theory because it means that it is not always attractive or desirable to keep on echoing that which has been said many times over. For reputational purposes one should *not* repeat others' viewpoints when those are well-worn. In sum, we propose that people who seek to maintain or secure a good reputation are most likely to speak out when public opinion is divided: it is at this time that, by joining the debate and giving voice to a distinctive viewpoint would make a difference in reputational terms.

We also believe that the focus on negative motives of fear and self-preservation is too one-sided: humans do not just form groups to avoid punishment. Indeed, there is abundant evidence that humans are intrinsically motivated to be prosocial. Especially relevant for the present paper is that people will go out of their way to maintain consensus and thereby preserve unity within their social circles [16, 33]. Maintenance of consensus can be a key way of preserving good social relations in a group [34]. Conversely, research shows that being in a group comes with an expectation of mutual agreement [35–37]. Thus, for reasons of maintaining social unity, it may be beneficial if group members echo previously aired views in order to validate them and reaffirm consensus [38, 39]. The other way around, disagreement and debate would be avoided by socially motivated group members in an attempt to preserve cohesion. In our view, then, echoing others' views may serve quite a different function to what is proposed by Spiral of Silence theory: to affirm consensus and thereby conserve unity [40].

This desire to maintain consensus applies especially to those who value cohesion and to those who are highly identified with the group [41, 42]. These are essentially collective concerns about the preservation of unity of the group or society. Those who are most concerned for the collective may therefore unwittingly contribute to this discrepancy of privately held views and public opinion.

In line with Spiral of Silence theory, we assume that individuals' attitudes may be more or less constant but their willingness to express changes depending on their perceptions of the public opinion [10, 21]. Going beyond Spiral of Silence theory, we propose that inferences about what public opinion is are based largely on the opinions that people can sample from those around them, from their inner circle as it were. Moreover, whether or not they choose to express themselves is mainly dependent on two distinct motives: (a) an individualistic motive to maintain or gain a good reputation and status within one's social circle, and (b) a collective or pro-social motive to maintain consensus within one's social circle. Of course such motives are not mutually exclusive and they may co-occur within the same person, but in the ABM presented below we shall assume that an agent has either one or the other.

We assume that those who individualistically seek to acquire a good reputation (reputation seekers) do so by airing their views when those views are both distinctive and likely to be supported—in other words when they can count on the support of a minority but they do not simply restate that which everyone says. With respect to pro-socially motivated people who seek to maintain group unity (consensus seekers), we assume that they will do so by airing their views when these are aligned with what the majority publicly. Because societies diverge in the extent to which each of these two motives is encouraged, and thus, likely to occur (e.g., [43]), we are interested in seeing how varying the relative number of reputation seekers and consensus seekers affects dynamics of public opinion. Besides this, we vary the size of the system, to get an insight in the stability of these dynamics when systems get larger.

## 1.4 Main aims and scope

The main aim of this paper is to model public opinion change as a function of small group dynamics which lead group members to either voice their attitude or remain silent.

These processes may be related to the opinions that individuals hear being expressed in their immediate environment (i.e., their in-group). As outlined above, we propose that two motives play a central role: pro-social motives to maintain consensus within one's group, and pro-self motives to maintain or gain good reputation within one's inner circle. In order to maintain consensus, group members are predicted to affirm the dominantly expressed opinion, voiced by a majority. In order for one's reputation to benefit, however, group members have to express an opinion that is a relative minority viewpoint. We assume that individuals

can only express opinions that are consistent with their privately held attitudes, or remain silent. By this we aim to examine not just what people say in small group discussions for social or selfish reasons, but also when they choose to remain silent for social or selfish reasons.

The goal of this study is to examine whether these group dynamics processes can produce a cyclical pattern of apparent public opinion fluctuation, which may occur without any actual attitude change. We examine whether it is possible to construct an agent-based model on the basis of our theoretical propositions, to query this model and to test our hypotheses. Holding the agents' private attitudes constant, we vary the size of the system and the relative numbers of reputation and consensus seekers within it.

## 2 The model

To describe and explain oscillations of publicly expressed opinion in conjunction with constancy of private attitudes, we developed and tested an agent-based model (ABM). The use of ABM provides unique advantages for the study of social phenomena as it enables the linking of distinct levels of analysis. Indeed, by using properties of single agents to predict emerging behaviors in their immediate social networks one can explain phenomena at a societal level [44]. By observing how the model behaves over time, the ABM approach allows us to check the consistency of our theoretical assumptions, and to examine results for unexpected and emergent properties [45]. Our model basically implements a Voter model [46], where each agent is surrounded by four neighbours (i.e., regular square lattice network [47]) and its behavior is influenced by the majority of these neighbours (i.e., Ising like evolution rule of behaviours [48]).

The following numerical recipe was adopted:

1. Random generation of $n$ agents with a probability $P_C = P_R = 0.5$ of being respectively consensus $A_i = C$ or reputation $A_i = R$ seekers.

2. Random allocation of these agents to the $N$ available spots within a square lattice network of dimension $\sqrt{n}$.

3. Random allocation of a Boolean attitude (i.e., $a$ or $b$) to each agent $i \in (1, n)$, independent of its population, with a probability of $P_a = P_b = 0.5$.

4. Random initialization (seeding) of the agents' behavior (i.e., silence or expression) vector $S_i^t \in [0, 1]$.

5. Evolution of the network following Table 1, for 10000 time steps storing the data of every single run.

6. Repetition of steps 1 to 5 for 10000 random permutations of different initial parameter configurations of the network.

**Table 1. The table shows the possible local network states, based on the behavior of the agents—Either silence ($S$) or expression ($E$)—And the population they belong to—Subdivided in consensus seekers ($C$) and reputation seekers ($R$).**

| Local network states | $C_M$ | $C_m$ | $R_M$ | $R_m$ |
|:---:|:---:|:---:|:---:|:---:|
| Total Silence | S | | E | |
| Stalemate | S | | E | |
| Majority with Minority | S | S | S | E |
| Partial Majority | E | S | S | S |
| Full Majority | E | S | S | S |

Agents $C$ and $R$ with a majority attitude have been labeled as $C_M$ and $R_M$, while agents with a minority attitude have been labeled as $C_m$ and $R_m$.

7. Repetition for different sizes of the network: 16, 36, 64, and 100 agents.

The model specifies three parameters that, when varied, determine the characteristics of the resulting network: the composition of the population, the agents' private attitudes and the state of the agents' local networks. In line with our theorizing (public opinion shifts without private attitude change), the agents' private attitudes are randomly assigned and held constant throughout the model simulations.

The model population consists of two different kinds of agents: "consensus seekers" and "reputation seekers" ($C_i$, and $R_i$ respectively). These are characterized by different decision rules for attitude expression as prescribed by our theory-based model visualized in Table 1 and outlined below. In short, $C_i$ acts on collective motives and $R_i$ acts on individualistic motives, as described before. $C$ and $R$ agents' decision to express their attitude (e) or remain silent (s) (their behavior), depends on the interaction between their own attitude (either $a$ or $b$) and the opinions expressed by their four neighbours (the *state* of their local network).

## 2.1 The state of local networks

Given a local network with four neighbours, when we represent an agent ($O_i$) as the center of a cross where the expressed opinions of its neighbours ($O_j$) are the arms, five possible local network states emerge, which can be summarized as follows (Fig 1):

- Total Silence (TS): all neighbors of agent $i$, at time $t$, do not express their attitude and thus remain silent (Eq 1).

$$TS_i^t = \begin{cases} 1, & \text{if} \quad \dfrac{\sum_{j=1}^{4} S_j^t}{4} = 1 \\ 0, & \text{if} \quad \dfrac{\sum_{j=1}^{4} S_j^t}{4} < 1 \end{cases} \tag{1}$$

- Stalemate (ST): half of the $n$ neighbours of the agent $i$ which express their opinion ($j$) at a certain time step ($t$) express opinion $a$ ($\sum_{j=1}^{4}[O_j^t = a] = \frac{n}{2}$), while the other half express opinion $b$ ($\sum_{j=1}^{4}[O_j^t = b] = \frac{n}{2}$). With [*Statement*] representing the Heaviside step function where [•] is equal to 1 whenever *Statement* is true, and 0 elsewhere.

$$ST_i^t = \begin{cases} 1, & \text{if} \quad \sum_{j=1}^{4}[O_j^t = a] = \sum_{j=1}^{4}[O_j^t = b] \\ 0, & \text{if} \quad \sum_{j=1}^{4}[O_j^t = a] \neq \sum_{j=1}^{4}[O_j^t = b] \end{cases} \tag{2}$$

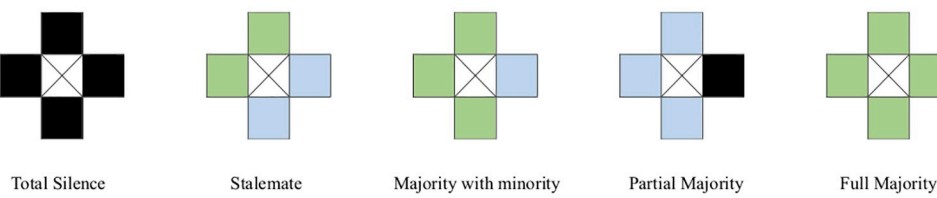

Total Silence          Stalemate          Majority with minority          Partial Majority          Full Majority

**Fig 1. Examples of the five local network states, defined by the attitude and the behavior of an agent's neighbours.** The black squares indicate the silent agents, while the green and blue squares represent the agents that express their opinion (either $a$ or $b$).

- Majority with Minority (MM): there is a majority expressing a certain opinion and a minority voicing the other opinion among the neighbours ($j$) of agent $i$. This can only occur if at least 3 neighbours express their opinion (Eq 3).

$$Mm_i^t = \begin{cases} 1, & \text{if } \sum_{j=1}^{4}[O_j^t = a] \neq \sum_{j=1}^{4}[O_j^t = b], \\ & \text{and } \sum_{j=1}^{4}[O_j^t = a] > 0 \\ & \text{and } \sum_{j=1}^{4}[O_j^t = b] > 0 \\ 0, & \text{Otherwise} \end{cases} \qquad (3)$$

- Partial Majority (PM): some neighbours express the same opinion, while the others remain silent (Eq 4).

$$PM_i^t = \begin{cases} 1, & \text{if } \sum_{j=1}^{4}[O_j^t = a] > 0 \\ & \text{and } \sum_{j=1}^{4}[O_j^t = b] = 0, \\ & \text{or } \sum_{j=1}^{4}[O_j^t = a] = 0 \\ & \text{and } \sum_{j=1}^{4}[O_j^t = b] > 0 \\ 0, & \text{Otherwise} \end{cases} \qquad (4)$$

- Full Majority (FM): all neighbours of agent $i$ express the same opinion (Eq 5).

$$FM_i^t = \begin{cases} 1, & \text{if } \sum_{j=1}^{4}[O_j^t = a] = 4 \\ & \text{and } \sum_{j=1}^{4}[O_j^t = b] = 0, \\ & \text{or } \sum_{j=1}^{4}[O_j^t = a] = 0 \\ & \text{and } \sum_{j=1}^{4}[O_j^t = b] = 4 \\ 0, & \text{Otherwise} \end{cases} \qquad (5)$$

## 2.2 Decision rules

The consequences of these different local network states for an agent depend on the population it belongs to. In the conditions *TS* and *ST*, an agent does not know whether there is a real majority in their group. That is why their attitude, if expressed, can be relevant to establish a majority. In this case, the *C* agents always remain silent $S_i = 1$, while the *R* agents always express their opinion $S_i = 0$. In the other three conditions (i.e., *Mm*, *PM*, *FM*), the behaviour of the agent is influenced by the agent's own attitude compared with the opinions expressed in its local network. A *C* agent holding a private attitude that is in line with the opinion unanimously expressed by its neighbours (i.e., a local majority ($C_M$) without a minority), will express its opinion. However, a *C* agent will remain silent whenever a local minority opinion is expressed, even if the agent's attitude matches the majority opinion. When the *C* agent has a minority attitude compared with the expressed local majority opinion $C_m$, the agent will also remain silent. From their side, *R* agents remain silent when their attitude is in line with the local

majority opinion $R_M$ (i.e., *PM*, *FM*, and *Mm*). When an *R* agent with a local minority attitude $R_m$ belongs to a network in which someone else already expresses this opinion (i.e., majority with minority), it will express its opinion.

## 2.3 Network dynamics

As outlined before, and in order to simulate the temporal evolution of the network, at each time step all agents in the network decided simultaneously whether to express their opinion or to remain silent, considering the interaction between their population, their own attitude and the state of their local network. The dynamics characterizing the network are the consequence of the parallel updating of the agents' state at each time step of the simulation. Specifically, an agent's probability of expressing is influenced by the state of its neighbours and a change in the agent's behavior would have an effect, in the following time step, on the state of its neighbours. Such a coupled mechanism makes the network self-correlated, and can present different scenarios at the equilibrium, depending on the initial conditions.

In particular, the network dynamics are always characterized by changes in the agents' behavior (i.e., expressing or silenced) at the beginning of each simulation. After a certain period, the network approaches an equilibrium which can be static (i.e., a fixed point) or dynamic (i.e., cyclical) in terms of the variations of one or more observables. In our case the observable of interest is the ratio between the two expressed opinions. The equilibrium can be a "simple" (i.e., static) equilibrium where the behaviour of each agent remains the same over time, or it can be more complex (i.e., dynamics) where agents continue to adjust their behavior over time.

An example of a complex equilibrium is the stable dynamical cycle, also called "limit cycle". A dynamical network can develop into a limit cycle when an invariant sequence of states (of arbitrary length) emerges that predicts all the subsequent steps of the network [49]. A limit cycle can be considered an equilibrium state of the network which is characterized by a period greater than 1. The emergence of limit cycles in the model is very interesting, because it suggests that the network may evolve to have stable periodical cycles akin to the oscillations of public opinion we aim to explain.

## 2.4 Observables

To test our hypotheses, we repeated, for different densities of R-agents, the model simulations with different initial conditions (i.e., agents' population, network configurations, attitude pattern, and initial expression) 10000 times, and we analyzed the distribution at the equilibrium of the following observables.

**2.4.1 Local networks.** Since each agent is "exposed" at each time step of the simulation to a certain configuration of neighbours' behaviors, the evolution of the system can be assessed considering the distribution of the number of agents presenting the five possible local network states emerging from the constraints of our model, e.g. Total Silence, Stalemate, Majority with minority, Partial Majority, Full Majority (as outlined in Fig 1), for different densities of *R* agents.

**2.4.2 Silence dynamics.** At every time step of each simulation, considering the last temporal window of length $\delta T$, the number of agents not expressing their opinion ($S_i^t = 1$) was summed up to define the average number of silent agents in the network $k$ ($\overline{S}_k^t$) for varying densities of *R* agents (Eq 6). We computed the average value across 10000 repetitions.

$$\overline{S}_k^t = \frac{\sum_{i=1}^n S_i^t}{n} \qquad (6)$$

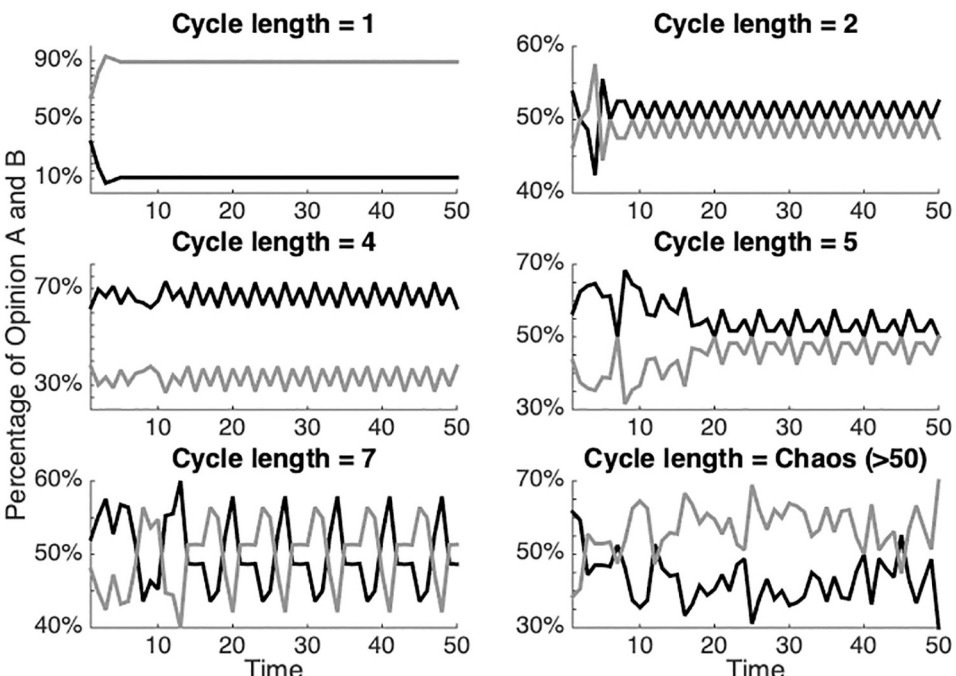

**Fig 2. Some representative examples of network equilibria with different cycle lengths, obtained by varying the number of reputation seekers ($R$).** Here it is possible to observe the oscillation of the expression of opinions $a$ (grey line) and $b$ (black line) over time.

**2.4.3 Average cycle length.** In order to assess the stability of the simulated system once it reaches an equilibrium, we defined an observable $\delta T$ to estimate the average cycle length: the number of time steps the network spent on average to return to the initial point of its stable trajectory in the phase space, as shown in Fig 2. The cycle length $\delta T$ was computed by considering the maximum value assumed by the distribution of the minimum temporal distances between two points on the systems' stable trajectory, just considering the second half of the simulation (when stable trajectories could have emerged). In other words, if the network assumes at time $t^1$ the state $X^*$, and returns to the same state $X^*$ at time $t^2$, the cycle length $\delta T$ is given by the equation $\delta T = t^2 - t^1 = 1$. As can be seen in Fig 5 the average cycle length shows a non-linear relationship with the percentage of reputation seekers ($R$) within the network, in particular the growth factor increases when the number of $R$ agents is increased, but only below a percentage of $R$ agents equal to 85%. Above such a sort of critical density of 85%, the average cycle length of the system starts to decrease when the number of reputation seekers is increased.

**2.4.4 Public-Private Opinion Discrepancy Index.** A key observable emerging from our simulations is the discrepancy between the publicly expressed opinions $\delta_E^t$ (Eq 7), and the private attitudes $\delta_A^t$ (Eq 8) of the agents at each time step.

$$\delta_E^t = \sum_{i=1}^n O_i(a)S_i^t - \sum_{j=1}^n O_j(b)S_j^t \tag{7}$$

Where $O_i(x)$ is the attitude of the subject $i$ that can be $a$ or $b$, and $S_i^t$ is the state of the agent $i$ at

a certain time step $t$.

$$\delta_A = \sum_{i=1}^{n} O_i(a) - \sum_{j=1}^{n} O_j(b) \tag{8}$$

The relation between $\delta_E$ and $\delta_A$ shows how closely publicly expressed opinions correspond with the distribution of privately held attitudes of the network. Publicly expressed opinions are completely aligned with privately held attitudes when $\delta_E = \delta_A$. But we expect that the network's parameters can combine to amplify or reduce the gap between $\delta_E$ and $\delta_A$, and we expect them to oscillate.

In order to investigate this dynamical feature of the network, we compare $\delta_E$ and $\delta_A$ by computing the "Public-Private Opinion Discrepancy" ($\theta$) Eq 9.

$$\theta = \frac{\delta_E^{Tmax-\tau:Tmax}}{\delta_A} \tag{9}$$

In this function $\tau$ is the final cycle length, *Tmax* is the end of the simulation, with $\theta \in [-\infty, \infty]$. Calculating $\theta$ in this way makes it possible to distinguish four possible scenario's of simulated networks:

- Correspondence: When $\theta$ is around 1, the equilibrium is characterized by correspondence between expressed opinions and private attitudes. In such cases $\delta_A = \overline{\delta_E}^{Tmax-\tau:Tmax}$.

- Amplified majority: When $\theta$ is greater than 1, the simulated network shows an expressed majority that is greater (in percentage) than the actual majority according to private attitudes. This means that minority opinions will be under-represented, so $|\overline{\delta_E}^{Tmax-\tau:Tmax})| > |\delta_A|$. If $sign(\overline{\delta_E}^{Tmax-\tau:Tmax}) = sign(\delta_A)$.

- Dampened majority: When $\theta$ is smaller than 1 but greater than 0, the simulated network presents a slight silencing of the majority and slight over-representation of the minority attitude in public expressions. The expressed majority is smaller than the actual majority according to private attitudes. Nevertheless, the expressed majority will still be larger than the expressed minority, so that $|\overline{\delta_E}^{Tmax-\tau:Tmax})| < |\delta_A|$, with $sign(\overline{\delta_E}^{Tmax-\tau:Tmax}) = sign(\delta_A)$. This means that the majority expressed opinion still accurately reflects what the actual majority is, according to private attitudes, but the minority viewpoint is over-represented.

- Silent Majority: Finally, when $\theta$ is smaller than 0, the simulated network is characterized by a silenced majority and an over-represented minority. In other words, the real and expressed majority are inverted: the majority according to the expressed opinions is actually the minority according to private attitudes, so that $sign(\overline{\delta_E}^{Tmax-\tau:Tmax}) \neq sign(\delta_A)$.

## 3 Computational results

This section describes the main results of the model simulations. By varying the density of $R$ agents in the network between $\in (0\%, 100\%)$ and running 10000 simulations per network configuration, we assessed the consequences of changing the relative proportion of the two populations in the network for the four observables: 1) local network states, 2) silence dynamics, 3) average cycle length, and 4) discrepancies between private attitude and public opinion.

Since the network size $N$ did not affect most of our results, the Fig 3 will always represent a network size $N$ of 100. In cases where network dynamics *do* change with the size of networks, we will report this.

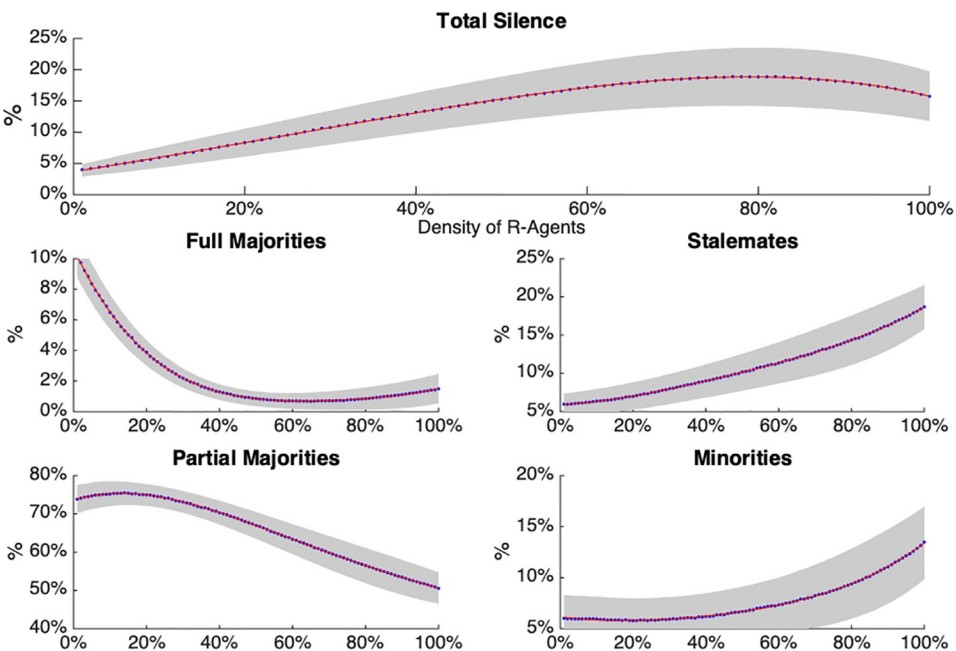

**Fig 3. The figure shows the relationship between the occurrence of the different local network states and the density of R agents.** In each graph, the average number of a given local network state—Stalemate, Minority, Partial Majority, Full Majority, Total Silence—is represented by blue dots, through which the red fitting line passes. The gray area around each line represents its standard deviation.

## 3.1 Local network states

The relationship between the occurrence of the different local network states and the density of *R* agents is shown in Fig 3. In general (i.e., across varying densities of *R*), the most commonly observed local network state is the Partial Majority (approximately ranging from 75% to 50%), which is followed by the Stalemates and Total Silence states (both ranging from approximately 5% to 20%), and by the vocal Minority states (approximately raging from 5% to 15%). The Full Majority states, where all neighbours voice an identical opinion, turns out to be the least common of these network states (approximately ranging from 8% to 0%).

Fig 3 shows that increasing the number of *R* agents leads to increasing occurrences of Stalemates and Majorities with Minorities, while the percentages of Partial Majorities and Full Majorities decrease. Finally the Total Silence states increased between *O*% and 80%, on average ranging within the interval (5%, 20%), and decreased for greater densities of *R* agents.

The probability of the different local network states appears to be related to *R* in a non-linear way. Thus, we can conclude that the more *R* (and the less *C*) agents a network contains, the more unstable local networks become in the sense that majorities become rarer and cases where opinions diverge within local networks become more common.

## 3.2 Silence dynamics

As explained before, agents decided to express their opinion or remain silent, based on the combination of the state of the agents' local network, their private attitude and whether they are seeking personal reputation or social consensus. The silence dynamics of the network are the aggregate result of this process.

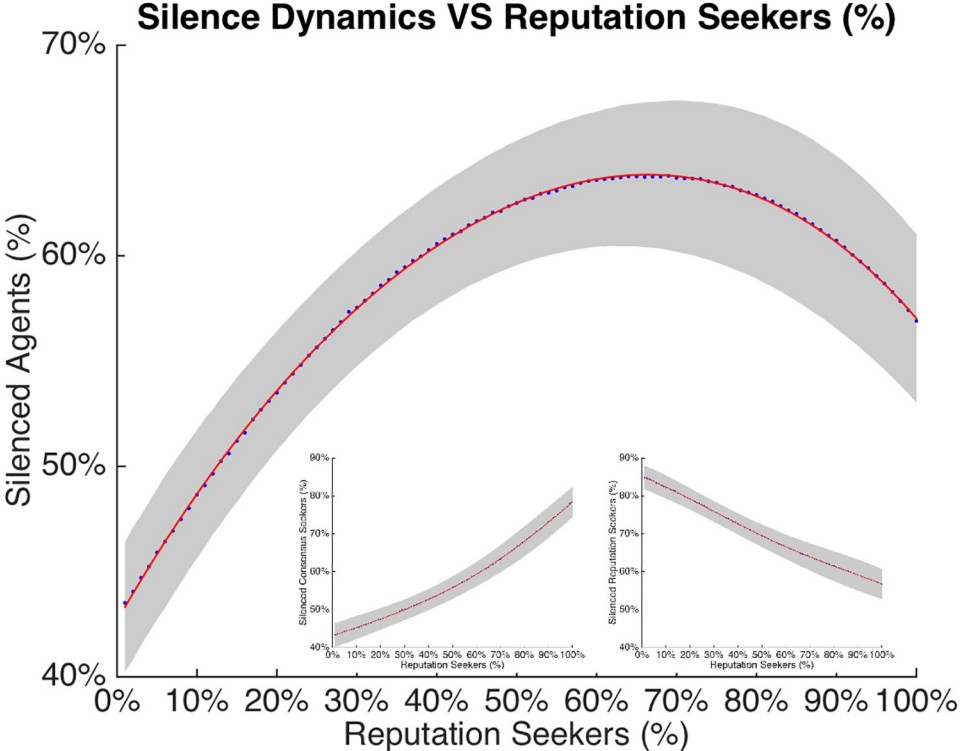

**Fig 4. The figure shows the silence density—Number of silent agents at the equilibrium as the density of *R* agents increases.** The blue dots indicate the average percentages of both parameters—silences and silent spots—for increasing percentages of *R*, with the best fitting line in red. The gray area around the points represents the standard deviation. As can be observed, the increase in the density of *R* agents seems to increase the number of silent agents. In particular, the highest proportion of silent agents occurs in networks which contain a percentage of *R* equal to 65%. The inserts represent the relation between percentage at the equilibrium of respectively Consensus Seekers (on the left), and Reputation Seekers (on the right), and the initial composition of the system.

The relation between the density of *R* agents and the average percentage of silent agents is reported in Fig 4. The density of silent agents shows a non-linear relationship with the density of *R* in the network. On average, the highest proportion of silent agents occurs in networks with a percentage of *R* equal to 65%. Interesting to note is that even though *R* agents in general express their opinion more frequently than consensus seeking agents do (see Table 1), increasing their density within the network from 0% to 65% appears to increase the number of silent agents (from 45% to 65%). This paradoxical result is due to the indirect influence that reputation seekers have on consensus seekers, as the latter choose to remain silent whenever there is disagreement in their local network. In other words, increasing the proportion of *R* agents enhances the likelihood that disagreement becomes apparent in expressed opinions within local networks, which reduces the likelihood that *C* agents express. The net effect of this is that simulated networks with more reputation seekers are characterized by an increase in silences.

### 3.3 Average cycle length

Results show a relation between the density of reputation seeking agents (*R*) and the average cycle length of the simulated network (Fig 5). When the density *R* is 0%, the simulated network always presents a static state (a final cycle length of 1). In other words, the network finds a stable solution in terms of the pattern of expressed opinions (i.e., the relative amount of *a* and *b*

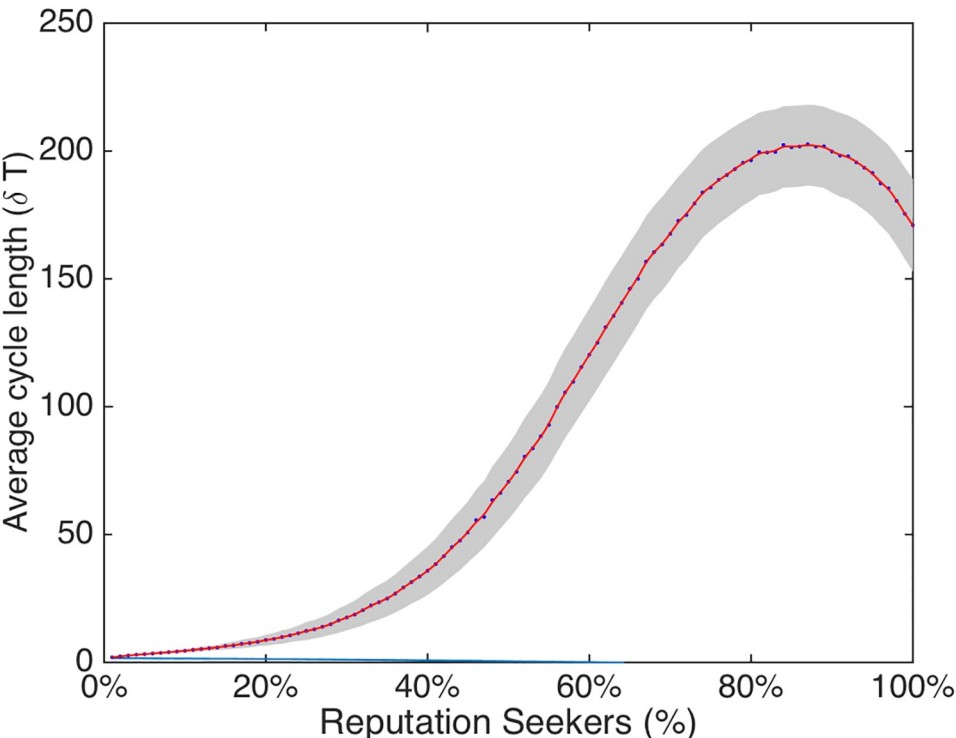

**Fig 5. The figure shows the relationship between the density of R agents and the average cycle length of the simulated network.** In red the spline interpolation represents the estimated mean of the average cycle length $\delta T$. The blue dots represent the average value and the gray zone indicates the standard deviation of $\delta T$. As can be observed, when the density of R increases, cycle lengths with periods greater than 1 become more common.

publicly expressed by the agents) and remains frozen, irrespective of the actual agents' attitudes. An example is provided in the upper-left corner of Fig 2.

As Fig 5 shows, when the density of *R* increases, cycle lengths with periods greater than 1 become more common. Here the simulated networks are characterized by dynamical loops: cycles of opinion expression of a certain length. These cycles are caused by the emergence of temporal patterns in local networks that influence agent's decisions to keep silent in certain rounds and to opine their attitude in other rounds.

Below a percentage of 20% *R*-agents, $\delta T$ appears to be ranging in the interval (2, 10). Above 20% of *R* a non-linear relationship between *R* density and $\delta T$ appears. Between a *R* density of 40% and 85% this is characterized by exponential growth in $\delta T$, whereas it is characterized by an exponential decrease between 85% and 100%. The average cycle length characterizing the simulated network assumes a maximum for a density of *R* agents around 85%. Such relation implies that the possible loops characterizing those equilibria of system we defined as "complex", would remains quite "short" for system characterized by small densities of reputation seekers. Nevertheless it would rapidly increasing whenever the density of *R* agents is greater than 20% and smaller than 85%, producing long temporal patterns (i.e., number of periodic public opinions fluctuations) characterizing the equilibria of the systems.

In sum, results show that once a simulation reaches an equilibrium, it is characterized by a cycle of a particular length with recurrent opinion dynamics. This is all due to majority and minority agents choosing to voice their opinion or to remain silent, depending on the state of their local network. As we saw in the previous sections, these local network states are more

commonly stalemates and minorities as the proportion of *R* agents increases. They are also characterized by many more silences, particularly among consensus seeking agents. This suggests that an increasing number of *R* agents means that the local networks less frequently have a clear-cut majority, which is associated with increasing cycle lengths and an increasing likelihood of chaotic (i.e., unpredictable) developments in expressed opinions.

## 3.4 Public-Private Opinion Discrepancies and emergent scenarios

As outlined above, taking into consideration the possible discrepancies between the expressed opinion ($\delta_E^t$) and private attitudes ($\delta_A$) within the simulated network, there are four possible scenarios that correspond to different value-ranges of $\theta$. These four scenarios are illustrated in Fig 6.

As shown in Fig 7, there is a non-linear relation between the density of *R* agents and both the minimum and the average values of $\theta$. As shown in the left-hand panel of Fig 7, the average value of $\theta$ is always greater than 0. This suggests that, on average, simulated networks in which a silent majority ($\theta < 0$) is found are relatively rare. When the system contains between 0 and 40% *R* agents, the most likely scenario is an amplified majority ($\theta > 1$). The scenario of correspondence ($\theta = 1$) between expressed opinions and actual attitudes was most probable when the percentage of *R* agents was around 50%. Finally, the dampened majority ($1 > \theta > 0$) was the most likely scenario when the density of *R* agents was above 60%.

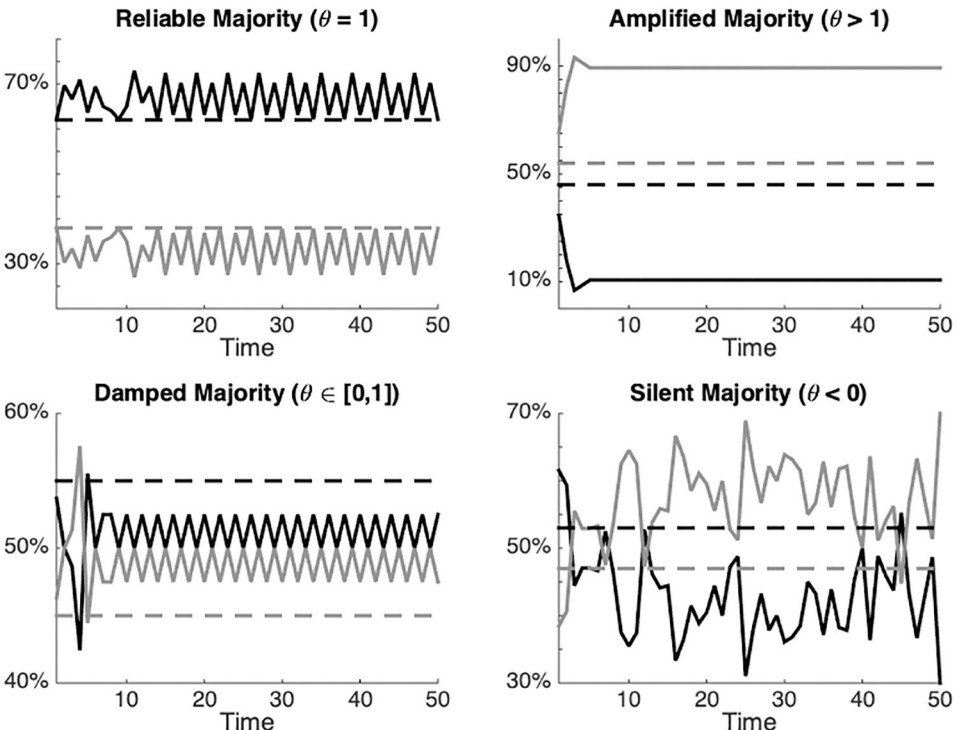

**Fig 6. Examples of the four possible scenarios found in the simulated networks when comparing expressed opinions with private attitudes.** Upper left: Correspondence between the expressed majority and the real majority. Upper right: Amplified majority where the expressed majority is greater than the real majority. Lower left: Dampened majority where the expressed majority is smaller than the real majority. Bottom right: Silent majority where the real minority is expressed as a majority viewpoint. The simulation was carried out with *N* = 100.

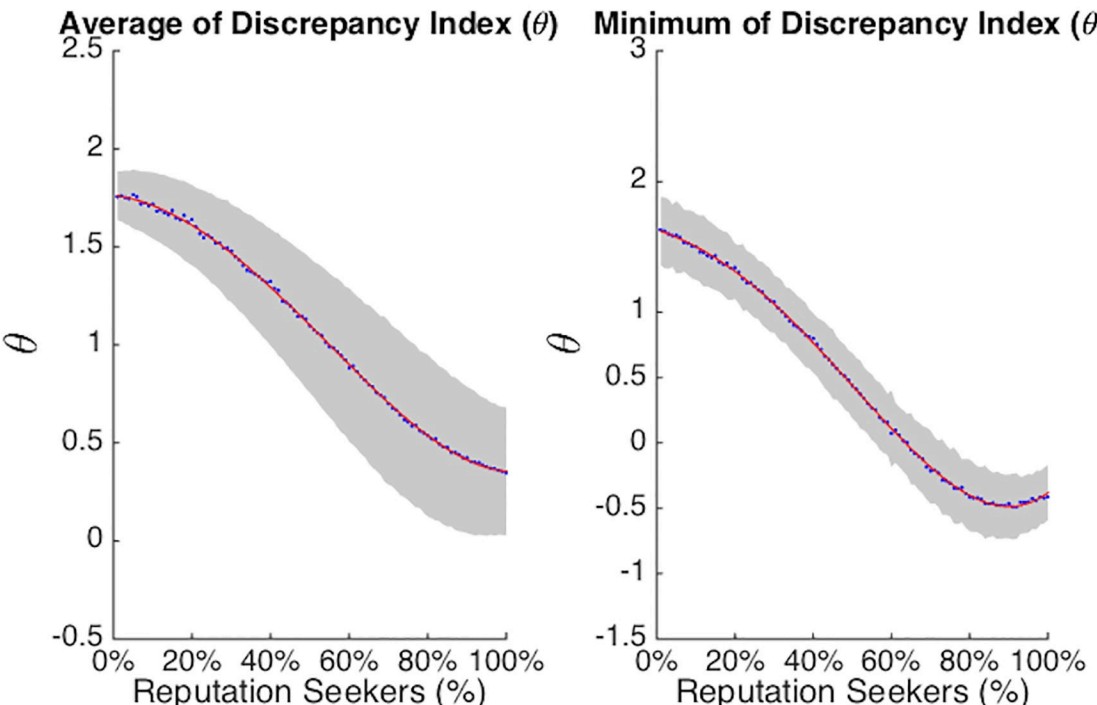

**Fig 7. The figure shows the averages (dots), the standard deviations (gray band), and the interpolations (red lines) describing the average and the minimum value of the Private-Public Opinion Discrepancy Index θ, for different densities of R agents.** In particular, depending on the density of R, we can observe an amplified majority ($\theta > 1$), a correspondence ($\theta = 1$), a dampened majority ($1 > \theta > 0$) or a silent majority scenario ($\theta < 0$).

As indicated by the gray band in the left-hand panel of Fig 7, the standard deviation of the average θ increased with an increasing density of R.

This phenomenon is explored more in-depth in the right-hand panel of Fig 7. As can be seen, there are numerous simulations in which the minimum of θ drops well below 0, when the density of R agents exceeds 50%. Thus, when the number of reputation seeking agents is well above half the population, situations in which the minority is over-represented become the norm and situations in which there is a silent majority become increasingly common. This is an interesting discovery in its own right: it suggests that minority over-representation can occur when, in local networks, divided opinions cause consensus seekers to remain silent and reputation seekers to speak up.

In sum, although the silent majority scenario never became the modal occurrence in the simulated networks, it did become more and more likely as the density of R agents exceeded 60%, with a maximum at a density of 85%.

In order to further explore these results, Fig 8 shows the percentage of occurrences of each scenario for four different network sizes: 100, 64, 36, and 16 agents. This figure illustrates clearly that, regardless of the size of the network, the density of R agents is associated with very different emergent scenarios of the simulated society as a whole.

Fig 8 shows that some parts of the distributions of scenario occurrences are unaffected by N. In particular, when the density of R agents is below 50%, most simulated networks converge on an amplified majority scenario. In other words, when the density of R is small, the minority tends to remain silent, leading to an overestimation of the majority size on the basis of

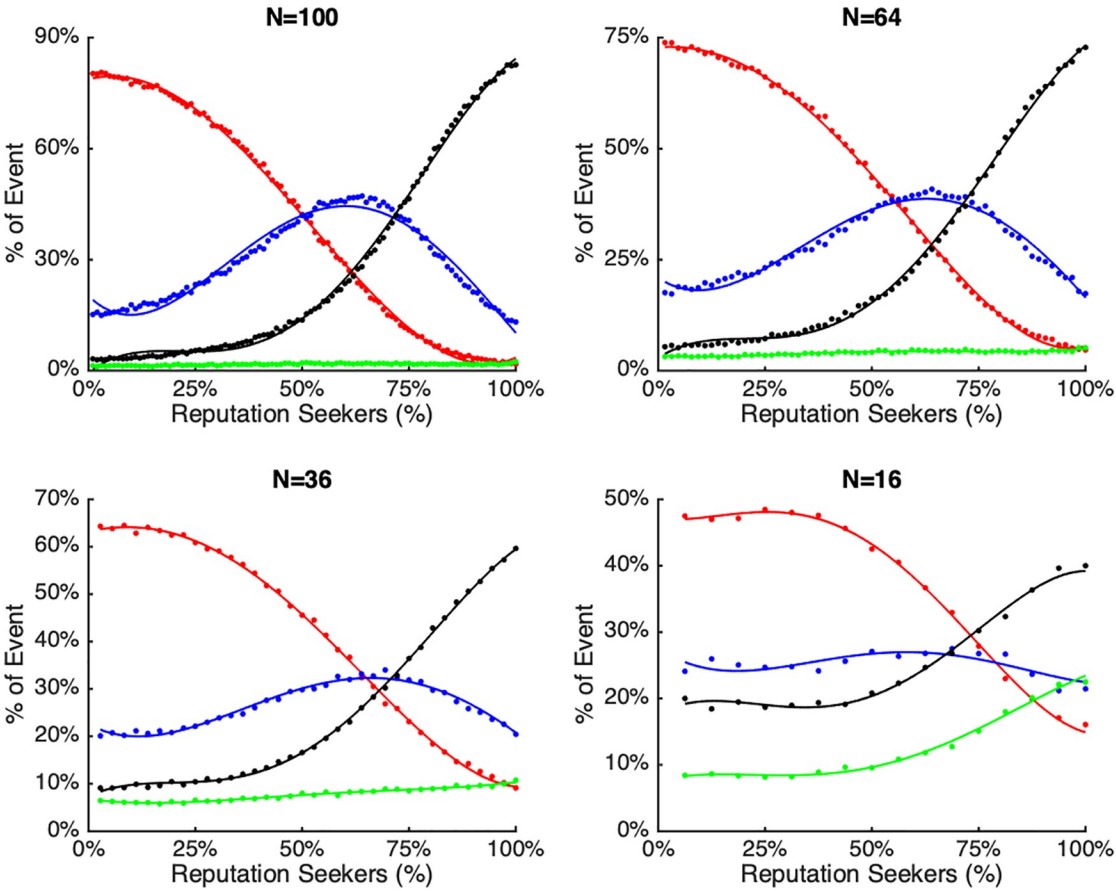

**Fig 8. The figure shows how the percentage of occurrence of each scenario (amplified majority in red, correspondence in blue, silent majority in green, dampened majority in black) varies in networks of different sizes ($N$ = 100, 64, 36, 16) and with different densities of R agents.** As can be observed, when the density of $R$ is below 50%, most of the networks converge on an amplified majority scenario, while when above 75%, most of the networks converge on a dampened majority scenario, where the minority is overestimated.

expressed opinions. The inference from this analysis is that network dynamics produce an intrinsic "suppression of dissent" when $R$ density is below 50%.

Conversely, when the density of $R$ agents is above 75%, most simulated networks converge on a dampened majority scenario. In other words, when the density of $R$ is high, the majority tends to remain silent leading to an overestimation of the number of agents holding the minority attitude. The inference is that network dynamics produce an intrinsic "accentuation of dissent" when $R$ density is above 75%.

A closer look at the four graphs reveals that the percentage of occurrences of each scenario changes with the size of the network. This reveals that in larger networks, the social structure of the simulated society is one in which over-representation of minority or majority opinions (at high or low densities of $R$, respectively) is the norm. In larger networks of $N$ = 100 and $N$ = 64, correspondence can also become the most frequently occurring scenario. This means that in a density of $R$ agents between 50% and 80% for the $N$ = 100 networks and in a smaller bandwidth for the other network sizes, it is more likely that scenarios emerge in which the minority and majority opinions are voiced in proportion to their actual representation within the population.

## 4 Discussion and interpretation

We proposed an agent-based model to examine how the micro-level dynamics of opinion expression affect the emerging macro-level dynamics of public opinion. In particular, we attempted to better understand the phenomenon that opinions that are publicly expressed on a certain topic can sometimes oscillate (occasionally quite strongly from pro to contra) whilst private attitudes remain much more stable. Results of the model simulations show that the proposed model can indeed account for these oscillations of public opinion over time.

With that, the current paper sheds new light not just on the phenomenon that sometimes, there can be a silent majority. It also sheds light on the phenomenon that sometimes the minority remains silent *and* on the processes by which societies can oscillate between these phenomena. In this discussion we shall consider the implications of the results of the model simulations.

### 4.1 Individual motives and small group dynamics

In line with classic approaches, such as Spiral of Silence theory, we distinguished between private attitudes and the expressed public opinion [24, 50]. People can remain silent, this previous work suggests, out of fear for the social repercussions of speaking out (e.g., social isolation or persecution). Prior agent-based models of these predictions show that such fears can, indeed, simulate the phenomenon of a silent majority.

The model in this paper advances our understanding of this process in two key ways. The first is theoretical: our model proposes that there are other motives than fear at work which play out as a function of micro-social dynamics in simulated social conversations. The second is that, as shown by the simulations, the interplay between individual motives and micro-social dynamics results in oscillations showing that spirals of silence can occur in which majorities *and minorities* can be silenced at times. The model informs us when particular silence phenomena are likely to dominate.

To elaborate, the main theoretical innovation of this work lies in a more accurate consideration of when and why people choose to speak out. The motives originally proposed by the Spiral of Silence theory [21] revolve around feelings of threat: people essentially conform to a felt social pressure to remain silent. And it is society at large (public opinion inferred for example from mass communications) which is responsible for this pressure.

Our model departs from earlier theories in three respects. First, it assumes that fear of rejection has a flip side: the desire to be successful and to achieve or maintain the respect of one's peers and in-group. In this respect, reputational considerations provide reasons for remaining silent, but also for speaking out. Second, it assumes that there are also prosocial motives at work, principally the desire to maintain group harmony by preserving consensus. Again, the motive to seek consensus can inform people when to remain silent and when to speak out. Finally, it assumes that the decision to speak out or remain silent is not made with reference to the community at large, but rather in relation to one's immediate environment: close others (friends, family, work colleagues and others whom we meet at a day to day basis) and their responses determine the immediate (relational) consequences of voicing one's opinion or remaining silent.

In sum, in comparison with previous approaches, the current model demonstrates that oscillations of silence can be explained a) at a different level: we based individuals' perceptions of public opinion on the expressed opinions of their inner circle (i.e., their direct neighbours) only, and b) with two distinct psycho-social motives: more individualistically motivated reputation seekers, who choose to voice their opinion or remain silent depending on what is required

to maintain or gain a good reputation in relation to proximate others, and consensus seekers, who choose voice or silence in order to maintain consensus within their inner circle (Table 1).

## 4.2 Emergent macro-level structures

In the model we examined whether the compositions of the population in terms of relative $R$ and $C$ agent density influenced the emergent macro-level properties of the network in terms of: (I) silence dynamics, (II) the stability and characteristics of the emergent scenarios, (III) discrepancies between public opinion and private attitudes, and (IV) the overall structure of the emergent network.

**4.2.1 Silence dynamics.**   As expected by our a priori hypotheses, varying the density of $C$ and $R$ agents within the network altered the silence dynamics. The density of reputation seeking agents predicted a density of silent agents ranging from 45% to 65%. The density of silent agents showed a curvilinear relation with the density of $R$ agents within the network, with the maximum number of silences occurring for a $R$ density of 65%. Increasing the density of $R$ agents from 0% to 65% also increased the density of silent agents. This is paradoxical because $R$ agents tend to express their opinion more frequently than consensus seeking agents (a ratio of approximately 3 to 2). The explanation is that the introduction of reputation seekers affects the behavior of those who seek to maintain consensus: $R$ agents introduce dissenting voices in their local environment (more stalemates and minorities and less partial and full majorities), in response to which $C$ agents choose to remain silent. This shifting of decisions to voice opinions is a key finding because the interplay between attitudes and these decisions is the foundation for all subsequent emergent structures. Increasing the density of $R$ agents even further from 65% to 100%, decreased the number of silent agents to around 55%, which might be due to the disappearance of consensus seekers from the network.

**4.2.2 Emergent temporal patterns (cycles).**   A further interesting insight from the model simulations is the emergence of different equilibria characterized by different cycle lengths (i.e., temporal patterns). The relation between the cycle length and the density of $R$ agents appears as non-linear. Reputation seekers actually appear to, in some way, "amplify" the potential dissent in the network by expressing a minority opinion and by preventing $C$ agents from expressing their views.

Indeed, the results show that when simulated societies are entirely made up of $C$ agents, networks quite quickly converge on a stable state in which the opinions expressed are the same every round (i.e., cycle length is 1). But as more and more $R$ agents are introduced, cycle lengths become longer and the characteristic oscillations of public opinion are becoming ever more common. As the number of $R$ agents increases even further, cycle lengths above 50 (which are essentially chaotic) become more frequent.

**4.2.3 Public-Private Opinion Discrepancies and the emergent network.**   The decisions of agents whether to express their opinions in their micro-level local networks, create distinctly different emergent patterns of publicly expressed opinions at the macro-level. We focused on the discrepancy between publicly expressed opinions (which we assumed to be variable) and privately held attitudes (which we held constant, assuming they are more stable). The theoretical scenarios foreseen by our model (i.e., amplified majority, correspondence, dampened and silent majority), were confirmed by numerical simulations and captured by the Public-Private Opinion Discrepancy ($\theta$), as represented in Fig 8.

The average value of $\theta$ gives insight in the modal state of the emergent network. The average $\theta$ decreased in a non-linear way with an increasing density of $R$ agents, with a minimum at a density of 85%. When the density of $R$ agents exceeded 65%, the average $\theta$ decreased between 1 and 0. This means that in these simulated societies, the majority view is dampened and

minority opinions are over-represented. Looking at the minimum value of $\theta$ allows us to understand better at which point a novel scenario becomes possible and likely. These results showed that when the density of $R$ agents exceeded 65%, some of the simulations achieved an equilibrium in which $\theta$ is smaller than 0. Thus, when there are many reputation seekers, it can happen that the public majority is actually the minority in terms of private attitudes.

The network dynamics also showed that minorities and majorities were proportionally represented in publicly voiced opinions (i.e., when $\theta$ is around 1) only when the density of $R$ agents was around 65%. As a consequence, we infer that reputation seekers play a key role in avoiding the phenomenon that minorities are "muzzled".

Integrating these different findings, it appears that networks with very small numbers of $R$ agents are extremely stable but are characterized by stifled minorities. Conversely, networks with very small numbers of $C$ agents are quite unstable and are increasingly characterized by stifled majorities. Somewhere beyond halfway, there is an optimum with proportional representation of minorities and majorities in networks with both $R$ and $C$ agents. But this optimum is no panacea: these networks often see oscillations between majority and minority dominance in public opinion. Another noteworthy finding is that at this optimum, the abundant voicing of dissent by $R$ agents ensures that minority viewpoints are adequately expressed, but this is also the point at which the number of agents that are silenced is at its maximum (around 65% of the agents remain silent). Thus, the introduction of more reputation seekers in the network improves minority representation but at some point also reduces the reliability of the publicly expressed opinions in the network.

Notably, these simulation results echo prior research that suggests that the motivation for reputational gain should not be equated with a purely selfish concern. Research has shown that deviating from the normative opinions in the group can be beneficial for group success: those who voice unpopular views and who seek to promote change are often seeking to improve their group [28, 51, 52]. We accordingly observed that to find an optimal solution in society does not require only consensus seekers. In fact, the results suggest that we need to take into account the interaction between consensus seekers and reputation seekers. An optimal proportion of both leads to a society that is quite stable, in which there is the highest correspondence between public opinion and privately held attitudes *and* which still allows for a degree of social change and flexibility. Finally, our simulations showed that larger networks (e.g., $N = 100$) in many respects have similar characteristics to smaller networks (e.g., $N = 16$). Although the smaller networks produced somewhat more variable scenarios, beyond the threshold of $N > 20$ small group dynamics produced similar macro-social scenarios, regardless the size of the network. Interestingly, we observed that large networks typically have longer cycles (i.e., more oscillations) and display more extreme over-representations of minority or majority at the equilibrium. This appears to be a non-linear relationship: the larger the network, the longer the cycles, and the more extreme the possible over (or under) representations of the real personal attitudes, whenever expressed public opinions are considered.

In sum, our model simulations suggest that the more reputation seekers a society contains, the more stalemates and minorities, as well as silences, can be observed on a local (micro-) level. This seems to lead to more over-representations of minority standpoints on a societal (macro-) level. Conversely, the more consensus seekers there are, the more partial and full majorities can be expected on a micro-level. This connects to an over-representation of majorities (i.e., silent minorities) on a macro-level. A correspondence between private attitudes and public opinions on a macro-level, which is arguably the most desirable scenario for a democratic society, occurs when around 70% of the agents is motivated by gaining reputation. Interestingly, this tends to co-occur with an abundance of total silence states (i.e., where an entire local network is silent) on a micro-level. Thus, our model suggests that the ideal of a society

where the public opinion accurately reflects private attitudes, requires dissenting voices as well as silenced attitudes.

## 4.3 Practical implications

Currently, large-scale national and international surveys, such as the World Values Survey or the Eurobarometer Survey, focus on measures of private attitudes on timely topics of a wide variety. However, quite surprisingly, such attitude measures often do not correspond to trends in public opinion witnessed in societies, nor with voting patterns. One prominent example of this is the fact that public opinion in many Western societies has increasingly shifted towards anti-immigrant sentiments [4], whilst attitudes regarding immigrants overall have been remarkably stable over the last decades [5, 53], notwithstanding considerable variation in how attitudes differ from region to region and occasionally also across time [54]. This disconnect between public discourse and private attitudes is most striking in research which relates news content (i.e., one supposedly influential form of public discourse) to immigration attitudes. In line with our assumptions, this shows that immigration attitudes are only marginally related to news discourse overall, but that attitudes *do* change to the degree that news portrays individual immigrants: here, exposure to immigrants tends to *improve* attitudes [55]. This implies that it is not just important to research why publicly expressed opinions may fluctuate but also why private attitudes may fluctuate. Studying both in tandem may shed light on their relationship.

To explain this occasional divergence of surveyed attitudes and public opinion, our model distinguishes between private attitudes and opinions that are expressed publicly. We showed that even when the private attitudes of the agents are held constant, publicly expressed opinions can oscillate and (depending on the reputational concerns of individual actors) situations can occur in which minorities as well as majorities are silenced, at certain times. Accordingly, this suggests that in order to gain a more complete understanding of the relationship between attitudes that are privately held and those that are publicly expressed, (inter)national surveys and panel studies should include questions not just about the beliefs and attitudes that people hold, but also about their willingness to express these beliefs and attitudes to others they meet on a regular basis. In other words, we should focus on who says what under which social conditions, to get an insight in the discrepancies between what is publicly voiced and privately thought.

## 4.4 Future research

In order to further validate the model and to inform future modelling attempts, it is important to anchor the model and its parameters in a program of empirical research. First, it might be possible to assess longitudinally how attitudes and expressed opinions on particular contentious topics and issues co-evolve over time.

Doing this on a societal-level can be very interesting but may also be rather difficult in practice. Real societies are more complex with more diverse (nuanced) opinions and many more local networks. Our results suggest that in a bigger society (larger $N$) the opinion dynamics become more chaotic and thus unpredictable. However, it is also possible, more feasible and just as interesting to study such phenomena in micro-societies (such as the merging of two departments in an organization, or in similar classroom settings, or in online chat groups). Such studies of smaller societies offer more control and thus are likely to be more informative about the relationship between micro-dynamics and more macro-level effects. This could help us better understand the psychological motives for speaking out and remaining silent. Moreover, such empirical research in micro-societies might increase our insight into the function describing the non-linear effect of the density of group members' personal motives to strive

for consensus or to make themselves heard (i.e., the proportion of *C* agents and *R* agents) on the Private-Public Opinion Discrepancy Index. This would serve to validate the model and could inform future extensions of it.

There are also numerous interesting extensions of the model itself. One way to extent the model would be to include the influence of online communication on public opinion dynamics. The current model only applies to an offline society. However, nowadays, people share and discuss their opinions not only with those they meet face-to-face, but also (even concurrently) in many online environments. The addition of an extra layer representing the online network on top of the offline society, will make the model more similar to a real society. This online society will have different characteristics, such as wider ranging local networks. Another way to enrich the model is to give the agents memory which enables them to base their behavior not only on the current state of affairs within their local network but also on their memory of the opinion distribution within their society. Memory might be especially relevant for the behavior of reputation seekers, who will need to compare the current opinion climate to the past to know what opinion is up and coming. Moreover, the addition of memory will allows us to study opinion dynamics on a longer term.

The present model assumes that agents either give their opinion or remain silent. Of course in reality people can choose other avenues: they can lie or dissimulate opinions. For example, in the classic social psychological studies by Asch, participants in a social setting when confronted with others who gave incorrect answers to a simple question chose, in a minority of cases, to follow suit and knowingly provide incorrect answers themselves too [56]. Such actions are not always voluntary: expression of certain opinions can be unlawful and in certain autocratic regimes one may risk prosecution even for remaining silent. A future model extension could therefore be to include a broader range of behaviors, such as the possibility of lying. Of course various other model extensions can be devised to gather interesting additional insights and also to make the model more ecological. For ecological and explorative reasons it would be interesting to add noise and to change the network topology. There may be other theoretically informed questions one might have. To this end, we encourage colleagues who are interested to use our code.

## 4.5 Concluding thoughts

At times of political polarization, it can seem as if only the most opinionated are willing to voice their views. In such situations, the moderates who seek to "hold things together" by promoting consensus choose to remain silent. In political debates, this "silent majority" is often called upon as a strategic ally. As this rhetoric at times of political polarization illustrates, the fact that true public opinion can be used so flexibly and creatively only underlines the value of the contribution we are seeking to make. We believe that it is of crucial importance to better understand why people choose to speak up as well as why they choose to remain silent. The macro-level consequences of such decisions are a foundation for better understanding how public opinions are shaped.

## Supporting information

**S1 File.**
(ZIP)

## Acknowledgments

Prof. Franco Bagnoli for the precious revision of the paper and software.

## Author Contributions

**Conceptualization:** Ren Manfredi, Andrea Guazzini, Carla Anne Roos, Tom Postmes, Namkje Koudenburg.

**Data curation:** Ren Manfredi, Andrea Guazzini.

**Formal analysis:** Andrea Guazzini.

**Investigation:** Andrea Guazzini, Carla Anne Roos.

**Methodology:** Andrea Guazzini, Tom Postmes.

**Project administration:** Andrea Guazzini, Tom Postmes.

**Software:** Andrea Guazzini.

**Supervision:** Ren Manfredi, Carla Anne Roos, Tom Postmes, Namkje Koudenburg.

**Validation:** Andrea Guazzini, Tom Postmes.

**Visualization:** Andrea Guazzini.

**Writing – original draft:** Andrea Guazzini, Carla Anne Roos, Tom Postmes, Namkje Koudenburg.

**Writing – review & editing:** Ren Manfredi, Andrea Guazzini, Carla Anne Roos, Tom Postmes, Namkje Koudenburg.

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
