## [Decision Letter · Decision Letter 0]

2 Sep 2020

PONE-D-20-20756

Private-Public Opinion Discrepancy

PLOS ONE

Dear Dr. Guazzini,

Thank you for submitting your manuscript to PLOS ONE. After careful consideration, we feel that it has merit but does not fully meet PLOS ONE’s publication criteria as it currently stands. Therefore, we invite you to submit a revised version of the manuscript that addresses the points raised during the review process.

As you can see, both reviewers found the work interesting and technically sound, thus my decision is Minor Revision. However, especially referee 2 asks for a better motivation of some modeling choices and I agree that it could help strengthening the message of the paper. 

We look forward to receiving your revised manuscript.

Kind regards,

Sandro Meloni, Ph.D.

Academic Editor

PLOS ONE

Journal Requirements:

Reviewers' comments:

Reviewer's Responses to Questions

**Comments to the Author**

1. Is the manuscript technically sound, and do the data support the conclusions?

Reviewer #1: Yes

Reviewer #2: Partly

2. Has the statistical analysis been performed appropriately and rigorously? 

Reviewer #1: Yes

Reviewer #2: I Don't Know

3. Have the authors made all data underlying the findings in their manuscript fully available?

Reviewer #1: Yes

Reviewer #2: No

4. Is the manuscript presented in an intelligible fashion and written in standard English?

Reviewer #1: Yes

Reviewer #2: Yes

5. Review Comments to the Author

Reviewer #1: SUMMARY:

Authors have developed an agent-based model in which they studied public and private attitudes in micro-societies.

The model has two types of agents: those who seek good reputation (R) and those promoting good harmony by reaching consensus (C).

The difference between the agents is whether or not they voice their opinion or remain silent, based on the behavior in their network. They also define the state of local networks that define if the agent will express its opinion or not, taking into account if they are Consensus seekers or Reputation seekers.

Model is based on the Spiral of Silence theory. Authors propose that within-group processes of reputation are a key factor in voicing one's attitudes.

They want to study the phenomenon that a society oscillates between public opinions (pro and contra) even when the average of privately held attitudes change little over time. Instead of using just the aggregate behavior, which people do not

know, people discuss and asses with small circles of intimates and personal networks.

Authors show that public opinions can oscillate and minorities, as well as, majorities can be silenced.

PROBLEMS WITH THE MANUSCRIPT:

Maybe it's just a matter of this draft, but if not, I recommend putting Fig 4 and 5 inside the text and not at the end of the document. And if they are at the end, they should be numbered so the reader can identify them.

QUESTIONS / COMMENTS TO THE AUTHORS:

Can individuals express opinions that aren't consistent with their privately held attitudes, instead of (selfishly) remain silent?

Have authors considered introducing noise and have a rough idea how the system will behave if introduced?

Authors state that "when there are many reputation seekers, it can happen that the public majority is actually the minority

in terms of private attitudes." Can this situation be overturned and the public opinion will change to the private one?

Authors have stated they would like to run experiments to test their model, but have they found empirical evidence from other authors that support the model?

Have the authors considered setting different topologies for the networks the agents interact in? If so, could they have a better impact on "common benefits"?

Have authors considered other types of agents (instead of just C and R) with different dynamics or the fact that Reputation seekers (Consensus seekers respectively) could change their behavior?

Is the code for the model available?

Reviewer #2: Thank you for giving me the opportunity to read and review the paper entitled “Private-Public Opinion Discrepancy”, submitted to PlosOne. In this paper, the authors present an agent-based model based on social psychological theories to investigate discrepancies between public opinion and private attitudes.

I hope that the following considerations and suggestions can help the authors to strengthen their manuscript:

1. The authors propose two motivations for agents to either remain silent or to voice their attitude. At times, the authors seem to suggest that their model provides insights into these motivations (e.g., line 141-147). However, the obtained results are rather descriptive and informative with regard to cyclical patterns of public opinion fluctuation and the discrepancy between private and public opinion on the macro level. Behavior that may be determined by striving for consensus or reputation on the other hand is determined by the model characteristics. In fact, I wonder whether the proposed pattern of behavior (Table 1) could not be explained by other motivations as well. Either way, while setting the stage and interpreting the findings it would be helpful to clearly distinguish between assumptions and potential explanations and warranted conclusions.

2. I highly appreciate the choice of method that the authors employed. As it is less commonly used within social psychological research – the field in which the study is theoretically rooted – it would be helpful to see a short justification for this choice.

3. The authors decided to vary the size of the system and the relative numbers of reputation and consensus seekers; however, no rationale explaining this decision is provided: From the theoretical background, it is unclear why these two factors were selected to vary.

4. With regard to future research, the authors could add some thoughts regarding potential downstream consequences on personally held attitudes for which the model currently does not assume any changes (line 167).

5. I could not find any information about the software used for the modelling and whether a script is available to allow replication.

6. Minor comments: In the abstract, it should be “rising concerns” and “however, statistics on private attitudes paint a different picture”. Moreover, I recommend substituting the word “parrot” (e.g., line 108) with a less evaluative word.

Let me conclude by saying that the topic of this paper is of great importance and the authors are to be commended for their work. I hope they will use these suggestions as they continue their work on this project.

6. PLOS authors have the option to publish the peer review history of their article (what does this mean?). If published, this will include your full peer review and any attached files.

Reviewer #1: **Yes: **Pablo Lozano

Reviewer #2: No

---

## [Author Response · Author response to Decision Letter 0]

1 Oct 2020

Dear Dr. Meloni,

We would like to thank you for the invitation to revise and resubmit our paper “Private-Public Opinion Discrepancy” to PLOS ONE. We are very grateful for the constructive and useful feedback provided by you and the two reviewers. The comments have helped us to substantially improve the paper. The most important changes concern the justification of the modeling choices made. We also included the simulation code underlying all the data in the supplementary materials.

We look forward to hearing your response. 

Kind regards,

The authors

Reviewer #1: 

PROBLEMS WITH THE MANUSCRIPT:

Maybe it's just a matter of this draft, but if not, I recommend putting Fig 4 and 5 inside the text and not at the end of the document. And if they are at the end, they should be numbered so the reader can identify them.

This is indeed due to the PLOS ONE submission requirements that state that figures have to be removed from the draft before submission.

QUESTIONS / COMMENTS TO THE AUTHORS:

Can individuals express opinions that aren't consistent with their privately held attitudes, instead of (selfishly) remain silent?

Thank you for your interesting suggestions. We think doing all of this (introducing lying, noice, and other typologies) might yield interesting novel insights. But for the moment we can not yet see a coherent theoretical viewpoint which would justify and necessitate us to take these next steps ourselves. We therefore invite other researchers to use our code to extend the model in these or other ways. We therefore included the following text at the end of paragraph 4.4: 

The present model assumes that agents either give their opinion or remain silent. Of course in reality people can choose other avenues: they can lie or dissimulate opinions. For example, in the classic social psychological studies by Asch, participants in a social setting when confronted with others who gave incorrect answers to a simple question chose, in a minority of cases, to follow suit and knowingly provide incorrect answers themselves too (Jetten & Hornsey, 2012). Such actions are not always voluntary: expression of certain opinions can be unlawful and in certain autocratic regimes one may risk prosecution even for remaining silent. A future model extension could therefore be to include a broader range of behaviors, such as the possibility of lying. Of course various other model extensions can be devised to gather interesting additional insights and also to make the model more ecological. For ecological and explorative reasons it would be interesting to add noise and to change the network topology. There may be other theoretically informed questions one might have. To this end, we encourage colleagues who are interested to use our code. (lines 700-713)

Have authors considered introducing noise and have a rough idea how the system will behave if introduced?

See point 1.

Authors state that "when there are many reputation seekers, it can happen that the public majority is actually the minority in terms of private attitudes." Can this situation be overturned and the public opinion will change to the private one?

See Reviewer 2, point 4.

Authors have stated they would like to run experiments to test their model, but have they found empirical evidence from other authors that support the model?

Of course there are many studies which show that group dynamics, needs for inclusion or desire for social approval influence people in the opinions they express (e.g., the classic Asch paradigm). But we do not know of any research that examines the relationship between the two motives central to this study and the combination of private attitudes and publicly expressed opinions in the longer run: the dynamics shown in this paper have no empirical parallel that we know of. Interestingly, we do not believe that the decision to remain silent has itself been investigated in empirical research much. Including this option may lead to quite interesting results.

Have the authors considered setting different topologies for the networks the agents interact in? If so, could they have a better impact on "common benefits"?

See point 1.

Have authors considered other types of agents (instead of just C and R) with different dynamics or the fact that Reputation seekers (Consensus seekers respectively) could change their behavior?

Yes, we have considered other agents (or rather motives) as mentioned in the introduction at the start of paragraph 1.3, lines 80-83. But in the present paper we rely on the very familiar prosocial - proself (or group - individual) distinction. We have not tested any other agents yet. 

With respect to changing behavior: see Reviewer 2, point 4.

Is the code for the model available? 

Yes, we now include the simulation code in the supplementary material.

Reviewer #2: 

The authors propose two motivations for agents to either remain silent or to voice their attitude. At times, the authors seem to suggest that their model provides insights into these motivations (e.g., line 141-147). However, the obtained results are rather descriptive and informative with regard to cyclical patterns of public opinion fluctuation and the discrepancy between private and public opinion on the macro level. Behavior that may be determined by striving for consensus or reputation on the other hand is determined by the model characteristics. In fact, I wonder whether the proposed pattern of behavior (Table 1) could not be explained by other motivations as well. Either way, while setting the stage and interpreting the findings it would be helpful to clearly distinguish between assumptions and potential explanations and warranted conclusions.

The reviewer’s point includes two questions:

a. Whether the proposed pattern of behavior could be explained by other motivations.

See Reviewer 1, point 6.

b. Whether we can clearly distinguish between assumptions and explanations and conclusions.

To address this point, we carefully read the manuscript again, to check whether our language reflected the difference between assumptions and conclusions. Where this was unclear, we made explicit when we were talking about assumptions...

For instance, in the introduction:

“We assume that those who individualistically seek to acquire a good reputation (reputation seekers) do so by airing their views when those views are both distinctive and likely to be supported---in other words when they can count on the support of a minority but they do not simply restate that which everyone says. With respect to pro-socially motivated people who seek to maintain group unity (consensus seekers), we assume that they will do so by airing their views when these are aligned with what the majority publicly expresses.” (lines 141-147)

And in the discussion:

“Our model departs from earlier theories in three respects. First, it assumes that fear of rejection has a flip side: the desire to be successful and to achieve or maintain the respect of one’s peers and in-group. In this respect, reputational considerations provide reasons for remaining silent, but also for speaking out. Second, it assumes that there are also prosocial motives at work, principally the desire to maintain group harmony by preserving consensus.” (lines 510-515)

But also clearly state what the present research demonstrated, namely: how oscillations of silence can be explained:

“In sum, in comparison with previous approaches, the current model demonstrates that oscillations of silence can be explained a) at a different level: we based individuals' perceptions of public opinion on the expressed opinions of their inner circle (i.e., their direct neighbours) only, and b) with two distinct psycho-social motives: more individualistically motivated reputation seekers who choose to voice their opinion or remain silent depending on what is required to maintain or gain a good reputation in relation to proximate others, and consensus seekers who choose voice or silence in order to maintain consensus within their inner circle (Table 1).” (lines 521-528)

I highly appreciate the choice of method that the authors employed. As it is less commonly used within social psychological research – the field in which the study is theoretically rooted – it would be helpful to see a short justification for this choice. 

Thank you for emphasizing this point. We agree and included a short justification at the start of paragraph 2:

“The use of ABM provides unique advantages for the study of social phenomena as it enables the linking of distinct levels of analysis. Indeed, by using properties of single agents to predict emerging behaviors in their immediate social networks one can explain phenomena at a societal level (Vallacher, 2017). By observing how the model behaves over time, the ABM approach allows us to check the consistency of our theoretical assumptions, and to examine results for unexpected and emergent properties (Goldstone, 2005).” (lines 171-181)

The authors decided to vary the size of the system and the relative numbers of reputation and consensus seekers; however, no rationale explaining this decision is provided: From the theoretical background, it is unclear why these two factors were selected to vary.

There is a methodological and theoretical rationale for varying these factors. From a methodological point of view, it is a logical first step in testing the model. For complex systems research it’s common practice to study how systems stabilize and behave by varying the size of the system and the other control parameters. This allows us to validate the model’s stability and to better understand its properties. Concerning the theoretical rationale for varying these factors, we now explain this in two sentences at the end of paragraph 1.3:

“Because societies diverge in the extent to which each of these two motives is encouraged, and thus, likely to occur (e.g., Hofstede, 1980), we are interested in seeing how varying the relative number of reputation seekers and consensus seekers affects dynamics of public opinion. Besides this, we vary the size of the system, to get an insight in the stability of these dynamics when systems get larger.” (lines 147-151)

With regard to future research, the authors could add some thoughts regarding potential downstream consequences on personally held attitudes for which the model currently does not assume any changes (line 167).

Thank you for this suggestion. Indeed, under specific conditions, attitudes can change as a result of being discussed (see for example the literature on persuasion). In fact, the current research provides clear support for a future research agenda that examines the dynamics between private attitudes and expressed opinions over time. We now make this explicit in paragraph 4.3:

“This disconnect between public discourse and private attitudes is most striking in research which relates news content (i.e., one supposedly influential form of public discourse) to immigration attitudes. In line with our assumptions, this shows that immigration attitudes are only marginally related to news discourse overall, but that attitudes do change to the degree that news portrays individual immigrants: here, exposure to immigrants tends to improve attitudes (Boomgaarden, 2009). This implies that it is not just important to research why publicly expressed opinions may fluctuate but also why private attitudes may fluctuate. Studying both in tandem may shed light on their relationship.” (lines 644 - 652)

I could not find any information about the software used for the modelling and whether a script is available to allow replication. 

See Reviewer 1, point 7.

Minor comments: In the abstract, it should be “rising concerns” and “however, statistics on private attitudes paint a different picture”. Moreover, I recommend substituting the word “parrot” (e.g., line 108) with a less evaluative word.

Thank you for pointing this out. We made the changes you suggested.

---

## [Decision Letter · Decision Letter 1]

28 Oct 2020

Private-Public Opinion Discrepancy

PONE-D-20-20756R1

Dear Dr. Guazzini,

We’re pleased to inform you that your manuscript has been judged scientifically suitable for publication and will be formally accepted for publication once it meets all outstanding technical requirements.

Kind regards,

Sandro Meloni, Ph.D.

Academic Editor

PLOS ONE

Additional Editor Comments (optional):

Reviewers' comments:

Reviewer's Responses to Questions

**Comments to the Author**

1. If the authors have adequately addressed your comments raised in a previous round of review and you feel that this manuscript is now acceptable for publication, you may indicate that here to bypass the “Comments to the Author” section, enter your conflict of interest statement in the “Confidential to Editor” section, and submit your "Accept" recommendation.

Reviewer #1: All comments have been addressed

Reviewer #2: All comments have been addressed

2. Is the manuscript technically sound, and do the data support the conclusions?

Reviewer #1: Yes

Reviewer #2: (No Response)

3. Has the statistical analysis been performed appropriately and rigorously? 

Reviewer #1: Yes

Reviewer #2: (No Response)

4. Have the authors made all data underlying the findings in their manuscript fully available?

Reviewer #1: Yes

Reviewer #2: (No Response)

5. Is the manuscript presented in an intelligible fashion and written in standard English?

Reviewer #1: Yes

Reviewer #2: (No Response)

6. Review Comments to the Author

Reviewer #1: Review and comments:

i) Figure 3 could be easier to interpret if the y-axis also has an axis title and not just "%" when future readers go through the article.

ii) Have the authors considered what is known now as "internet trolls" when they say: "Especially relevant for the present paper is that people will go out of their way to maintain consensus and thereby preserve unity within their social circles."? Where those individuals try to disrupt the

iii) Have authors tried to model agents with the possibility of having both motives (a) an individualistic motive to maintain or gain a good reputation and status within one’s social circle, and (b) a collective or pro-social motive to maintain consensus within one’s social circle? They state only one motive is possible, but it was a simplification or they tried the model with both and didn't obtain any useful result?

Reviewer #2: (No Response)

7. PLOS authors have the option to publish the peer review history of their article (what does this mean?). If published, this will include your full peer review and any attached files.

Reviewer #1: **Yes: **Pablo Lozano

Reviewer #2: No

---

## [Editor Report · Acceptance letter]

30 Oct 2020

PONE-D-20-20756R1 

Private-Public Opinion Discrepancy 

Dear Dr. Guazzini:

I'm pleased to inform you that your manuscript has been deemed suitable for publication in PLOS ONE. Congratulations! Your manuscript is now with our production department. 

Kind regards, 

on behalf of

Dr. Sandro Meloni 

Academic Editor

PLOS ONE